# Iron-Sepiolite High-Performance Magnetorheological Polishing Fluid with Reduced Sedimentation

**DOI:** 10.3390/ijms232012187

**Published:** 2022-10-13

**Authors:** Radoslav Milde, Robert Moucka, Michal Sedlacik, Vladimir Pata

**Affiliations:** 1Department of Production Engineering, Faculty of Technology, Tomas Bata University in Zlín, Vavreckova 275, 760 01 Zlin, Czech Republic; 2Polymer Centre, Faculty of Technology, Tomas Bata University in Zlín, Vavreckova 275, 760 01 Zlin, Czech Republic; 3Centre of Polymer Systems, University Institute, Tomas Bata University in Zlín, Trida T. Bati 5678, 760 01 Zlin, Czech Republic

**Keywords:** polishing, magnetorheology, sedimentation, slurry, clay, 3D texture

## Abstract

A sedimentation-stable magnetorheological (MR) polishing slurry on the basis of ferrofluid, iron particles, Al_2_O_3,_ and clay nanofiller in the form of sepiolite intended for MR polishing has been designed, prepared, and its polishing efficiency verified. Added clay substantially improved sedimentation stability of the slurry, decreasing its sedimentation rate to a quarter of its original value (1.8 to 0.45 mg s^−1^) while otherwise maintaining its good abrasive properties. The magnetisation curve measurement proved that designed slurry is soft magnetic material with no hysteresis, and its further suitability for MR polishing was confirmed by its magnetorheology namely in the quadratically increased yield stress due to the effect of applied magnetic field (0 to 600 kA m^−1^). The efficiency of the MR polishing process was tested on the flat samples of injection-moulded polyamide and verified by surface roughness/3D texture measurement. The resulting new composition of the MR polishing slurry exhibits a long-term stable system with a wide application window in the MR polishing process.

## 1. Introduction

Magnetorheological (MR) surface polishing is an advanced finishing method capable of achieving very low micro-roughness and high surface accuracy [1]. Benefits of the technique comprise no wear of the polishing tool as it is constantly recreated in the form of solidified MR polishing fluid under the effect of an external magnetic field in the process known as an MR effect [2]. In this, upon application of an external magnetic field, magnetic particles form aggregates thus qualitatively changing rheological properties of the suspension, i.e., yield stress, apparent shear viscosity, and elastic modulus, consequently turning its deformation behaviour from liquid to solid-like state. If the intensity of the magnetic field is further increased, formed internal chain-like structures are compacted and solidify [3]. The stiffness of these internal structures is proportional to the intensity of the applied field. The whole process of alignment usually lasts in the order of milliseconds, and after the removal of the external magnetic field the system immediately returns to its original liquid state with typically Newtonian type behaviour [4]. The described MR effect is employed in various industries, from automotive, to construction, to military, in the form of various damping systems or small mechanical components [5].

Another key segment where MR effect is employed is MR finishing [6]. Finishing is a key technology operation which heavily impacts final surface properties, which in turn influences either functional properties or contact place with surroundings. In MR surface polishing, which is one of the finishing techniques, the machined surface of the sample is in contact with MR suspension containing also abrasive particles such as corundum [7], cubic boron nitride [8], etc., also known as a slurry. Among the advantages of this technology is the minimisation of surface damage to the workpiece and application to different types of materials, such as polymers [9], optical glasses [10], etc. Importantly the technology is applicable to both planar and spherical surfaces [11,12,13,14].

Depending on the construction details of the device, the variable parameters during the MR surface polishing process affecting the material removal rates can generally be summarised as translation oscillation, work gap, period ratio, eccentricity, wheel speed, intensity of the magnetic field generated, and composition of MR slurry [15].

In general, the MR slurry consists of magnetic particles, a carrier liquid, and abrasive grains. Magnetic particles are a key component of the slurry as they facilitate the interaction of the slurry with the external magnetic field; spherical (carbonyl) iron particles (IP) with a size of 0.5–10 μm are often used [16,17]. Magnetic particles need to have sufficient magnetic permeability, which increases with cumulative particle size, however, the greater the particle size the quicker they sediment.

Abrasive particles are another essential component of MR slurry for the MR surface polishing process. As mentioned, during the polishing process, the magnetic particles form chain-like structures in the direction of the external magnetic field, while the abrasive particles are pressed on the surface of the workpiece [1]. It has been shown that the process of removing MR material by polishing is a result of the penetration of abrasive particles into the workpiece surface. The resulting surface roughness of the workpiece is primarily determined by the size of the abrasive particles and the force acting on the abrasive particles [18].

The role of the carrier fluid in the MR slurry is to give it its liquid character. Due to the low viscosity, water or ferrofluid, which is an analogue to MR suspensions differing only in the size of the dispersed particles, are commonly used as a carrier liquid [4].

There are several issues related to MR polishing/slurry, such as oxidation stability of the magnetic particles, which are exposed to high temperatures and/or the presence of reactive substances during operation [19], however, arguably the most serious is the sedimentation of the magnetic particles [2]. High density difference between magnetic particles (iron) and carrier liquid (water) leads to a very intense sedimentation process, which undesirably affects the effectivity of the MR surface polishing process.

To decrease sedimentation, various strategies were explored in the past, one of the most common strategies proved to be magnetic particles of the core-shell type [20]. A magnetic particle (the core) is, in this approach, coated by a polymeric layer (the shell). The shell generally has a lower density (polymers) compared to the metal magnetic cores. Therefore, its presence positively affects the sedimentation rate of the particles in the MR system [11]. Another benefit of using polymer coatings (polyaniline [21], polystyrene [22], polysiloxane [3] or others) on the particles is the increase in their affinity to the carrier liquid, which again prolongs stability of the MR system. Organic packaging also typically reduces wear and tear on the surface of the equipment and extends the life of the machine, which is important for most applications. Although being a valid way of dealing with the challenge of particles’ sedimentation, it also introduces new issues. Firstly, the lengthy synthesis of composite particles of the core-shell type prolongs the slurry preparation as it requires additional synthesis step(s). Secondly, the coating procedure itself is often not eco-friendly due to the usage of certain dangerous solvents (tetrahydrofuran, toluene, etc.). Lastly, the coating layer suppresses magnetic properties of the filler rendering the system less effective. Although this issue can be resolved employing ATRP polymerization [23,24], it is mostly limited to the lab because scale-up for industrial use is complicated. Other approaches to slow down sedimentation of MR suspensions include adding an additive such as nanoparticles of carbon to the system, which plays a role of steric barrier hindering the sedimentation of iron microparticles resulting in system stability [25], the use of shear thickening agents such as fumed silica [8,26], or using deflocculated MR polishing slurry [27]. However, these approaches insert another component into the system’s composition which can reduce its sensitivity to the magnetic field and potentially increase the viscosity of the system in its inactive state (without an applied magnetic field), which can be a limiting factor in some applications, among them MR polishing due to the need to transport MR slurry during the process. Another way of hampering (or at least considerably impeding sedimentation of magnetic particles) can be realised through the formation of a gel structure in the slurry. In a recent study, it was shown that the presence of clay particles in silicone oil initializes the formation of a gel structure that hampers sedimentation of the dispersed particles [28,29].

In this study, the influence of a low amount of clay nanofiller in the form of sepiolite on the process of MR surface polishing of PA-6, and on the long-term stability of the prepared MR slurry, will be investigated. The response of the MR slurry to an external magnetic field of the corresponding properties used in the MR polishing process will be investigated through MR measurement using a rotational rheometer which will simulate the actual processing window used in the MR polishing process. Magnetorheological polishing will be carried out on a newly developed device for polishing planar non-metallic surfaces with subsequent evaluation of surface treatments using 3D texture analysis.

## 2. Results and Discussion

### 2.1. Particles Morphology

Each of the three main fillers has a different morphology as revealed by scanning electron microscopy (SEM). While magnetic IP are almost perfect spheres of diameter ranging from 0.5 to 4 µm (Figure 1a), abrasive filler comes in 2D, flaky, sheet-like shapes about 1 µm thick and several micrometres wide (Figure 1b). Finally, clay additive forms needle-like (fibrous)morphology (Figure 1c). The SEM image of the final MR slurry is not presented for the reason of clarity where the concentration of IP is much higher and the other components could be overshadowed.

Table 1 represents the composition of MR slurry under investigation.

The individual components for MR slurry under investigation were selected from the literature in such a way to represent those most commonly used (in terms of material and concentration).

### 2.2. Magnetorheology

Rheological properties of ferrofluid and MR slurry without a magnetic field and magnetic fields of varying intensity (see legend) are shown in Figure 2, where also an area of polishing conditions (depending on used rotations per second (rps) and gap) is marked (orange rectangle). As can be seen, flow properties are dominated by the MR slurry (full symbols), which has several orders higher shear stress over the ferrofluid used as a carrier liquid (empty symbols). Application of magnetic field shifts the measured dependence up as the formation of internal chain-like structure formed by magnetized IP further prevents the flow. In order to extract yield stress, the flow curves were measured from relatively low (compared to operating conditions) shear rates, and although these low shear rates are feasible in terms of device capability, from the practical point of view such low shear rates would either lead to poor polishing results or require unproportionable prolongation of polishing time.

The ferrofluid itself has low shear stress, which is proportional to the viscosity, in the whole range of shear rates, and even though its shear stress increases with application of external magnetic field as a direct consequence of MR effect even at 420 kA m^−1^ it still does not reach values of MR highly loaded (presence of IP, Al_2_O_3_, clay) slurry in the off-state. Thus, the effect of loading MR slurry with filler clearly dominates over MR effect in the pure ferrofluid. Apart from providing a liquid character to the MR slurry, ferrofluid also contributes to the enhancement of long-term stability of the MR slurry (magnetic nanoparticles form a steric obstacle in the sedimentation trajectory of IP and Al_2_O_3_) as well as an intensification of the MR effect due to its magnetic character [30].

Magnetorheological slurry has nearly Newtonian character, however, with the application of a magnetic field this changes to pseudoplastic behaviour with pronounced yield stress. In the presence of a magnetic field, shear stress rises almost threefold owing to the MR effect (formation of internal chain-like structure). A plateau region present at low shear rates signifies a region where magnetic forces dominate over hydrodynamic ones. The fact that flow curves shift up (i.e., shear stress increases) with increasing intensity of magnetic field means that the inner chain-like structure does not completely break down, i.e., does not follow a field-off curve. Plateau in the first region of the flow curve is related to flow field driven disruption and nearly immediate restoring of chain-like structures [31]. It holds that the higher the intensity of the magnetic field the more robust chains are formed. Taking into account the shift of flow curves upwards with increasing magnetic field in the investigated intensity range, it can be concluded that the given system is not fully saturated, and, for the given MR slurry composition, more rigid structures could be achieved. Our processing conditions, i.e., rotation speed of the magnet and the gap (defining shear rate; orange box in Figure 2), are well in the investigated range of rheological measurements.

To evaluate the MR performance of MR slurry, the $\tau_{0}$ values taken as values of the shear stress at very low shear rates were plotted against the external magnetic field intensity (Figure 3). Evidently, the dynamic yield stress of the MR slurry was found to increase with the intensity of the applied magnetic field. This increase is very closely proportional to the square of the magnetic field intensity, $\tau_{0}\propto H^{2}$, (exact exponent being 2.14), which in the log–log plot gives linear dependence with the slope of 2. According to the finite-element analysis and linear magnetostatics, this result means that the applied magnetic field intensity is lower than the critical one, at which the contact or polar regions of each particle saturate [32]. In other words, further significant increase of $\tau_{0}$ with an increase of magnetic field intensity can be expected, and the MR slurry possesses high MR performance. However, $\tau_{0}$ of the MR slurry must always be adapted (by the magnetic field applied) to the character of the polished surface, when, for example, it would not be desirable for polymers to achieve a significant stiffness of the MR slurry (the surface would be rather scratched, see below) compared to the requirements for ceramics.

### 2.3. Magnetic Properties of Magnetorheological Slurry

Measurement of the magnetisation loop of the MR slurry shows practically no hysteresis, i.e., exhibits nearly zero remanent magnetisation and coercivity (Figure 4).

This means the MR slurry is an extremely soft magnetic. From mutual comparison of the measured magnetisation curve for the MR slurry and the values of magnetic field (384, 420 and 460 kA/m) used during the MR surface polishing process, it can be seen that we are close to—but not quite at—the saturation magnetisation, which is confirmed even by flow curves that have not reached a point where they would overlap.

### 2.4. Sedimentation

As abovementioned, the sedimentation represents one of the most important problems hindering wider utilization of MR systems with liquid carrier. For this reason, sepiolite clay mineral was added in low concentrations to the standard MR slurry to play the role of a thickening agent suppressing the filler sedimentation issue in this study. The sedimentation stability of MR fluids, of which MR slurries are a subset, is usually assessed by visual observation [33] or instrumental setup [34], e.g., Turbiscan, which measures the transmitted and the backscattered near-infrared monochromatic light passing through a sample using two synchronous optical sensors. However, this visual observation is not possible in the case of the MR slurry investigated in this study because the carrier liquid is a ferrofluid, which under standard conditions is a stable opaque system preventing any optical observation of the sedimentation of other components. Sedimentation rate of MR slurry was determined in this study through another kind of direct observation method for particle sedimentation based on the measurement of mass increase in time (Figure 5). After a certain time period (approx. 500 s), in which both processes reach equilibrium, mass increases steadily in an approximately linear fashion with time. A slope of this dependence (in mg s^−1^) was used to quantify the observed sedimentation rate. Thus, the original MR slurry (1.77 mg s^−1^) sediments 4× quicker than the one with clay added (0.45 mg s^−1^). This is in good agreement with previous studies finding a positive effect of sepiolite belonging to the group of typical natural materials on the sedimentation stability of other smart liquid systems [35]. The observed results of suppressed sedimentation of MR slurry are even better than, for example, previously published results for MR slurry containing core-shell structured IP [36]. Sedimentation of magnetic particles and abrasive particles in the real use of MR slurry in the MR polishing process can occur in devices that draw MR paste from a reservoir. In such a case, the currently needed MR slurry may be dosed with a lower concentration, which would ultimately lead to a less efficient polishing process, since the MR effect is dependent on the concentration of the magnetic filler [4].

### 2.5. Magnetorheological Polishing

The samples were subjected to an MR polishing test using the abovementioned slurry. Surface quality was assessed with a 3D microscope. The surface of the PA-6 test sample was polished at the following conditions: polishing time 5 min, polishing rate 15 rps, the gap between the magnet and the workpiece (0.5, 1.0, 1.5 mm). Prior to surface characterisation, the polished samples were sonicated in order to remove remnants of MR slurry from the samples’ recesses.

The surface was characterised by four basic parameters (also shown in the top left corner of the figures): an average arithmetic deviation of the investigated profile *S*_a_, maximum peak height *S*_p_, maximum valley height *S*_v_, and the sum of the maximum peak and maximum valley *S*_z_.

As can be seen in Figure 6, the polishing process has decreased both the average arithmetic deviation of the profile (*S*_a_) and the sum of maximum peak and maximum valley (*S*_z_) within the range of the basic length. These two parameters are good indicators of whether the polishing has or has not taken place.

However, exclusively on the basis of these two parameters this cannot be claimed with 100% certainty. Hence, auxiliary parameters *S*_p_ and *S*_v_ come into play. After polishing, the value of the *S*_v_ parameter is to remain approximately the same or to decrease slightly, while the *S*_p_ parameter should be diminished considerably. Comparing *S*_p_ before (Figure 6a) and after (Figure 6b) polishing, one can notice a 15% drop which indicates significant improvement of surface quality in terms of its smoothness, which is also obvious from 3D scans of the sample surface. Therefore, we can conclude that proposed MR slurry is suitable for polishing non-metallic surfaces.

On the basis of individual roughness parameters (*S*_a_, *S*_z_, *S*_v_, *S*_p_) evaluation, which has been carried out according to ISO 4288 and ISO 4287 standards, it can be concluded that there are statistically significant differences between initial roughness parameters and those measured after the polishing process (10 samples were measured under the same polishing conditions). From these results, relative differences were determined and plotted against the polishing gap (Figure 7). As can be seen from the regression analysis of the selected parameter *S*_p_, its relative difference, Δ*S*_p,norm_, linearly increases with the gap. *S*_p_ parameter was chosen as it is not affected by other influences, such as *S*_v_ which is sensitive to valleys filling up with MR slurry. This would suggest that the polishing process tends to yield better results when higher magnetic fields are applied (at least within the investigated range of 384 to 460 kA/m) as peaks decrease with it, which is in a good correlation with results observed in [36] for silica-coated IP-based MR polishing slurry. However, one needs to bear in mind that this is valid only in the case of comparable starting values of *S*_p_, i.e., samples with similar surface roughness. The used range of magnetic field intensity was chosen optimally, as can be seen from Figure 7, when at the magnetic field intensity of 460 kA/m (gap = 0.5 mm), the chain-like structures of the IP were too rigid and the surface of the workpiece was scratched rather than polished. Although higher Δ*S*_p,norm_ was achieved at a magnetic field intensity of 384 kA/m (gap = 1.5) compared to 420 kA/m (gap = 1.0 mm), a more pronounced dispersion of these values (wider 95% confidence interval) was observed at the same time. A change in the course of this curve (Figure 7) can be expected when the magnetic field intensity is further reduced (larger gap), because insufficiently rigid chain-like structures of IP and thus an ineffective polishing process will occur. Therefore, the value of 420 kA/m (gap = 1.0 mm) can be considered ideal and the surface characteristics for this value are presented in Figure 6. The quality of the surface could be further changed by changing other parameters, such as rotation speed or process time, but this will be part of a future study, in which not only the mentioned parameters will be included in the creation of a predictive model for evaluating the 3D texture of surfaces using a regression analysis and artificial intelligence techniques. The current study is focused on proving the higher sedimentation stability of the used MR slurry with confirmation of its usability in the MR polishing process (Figure 6 and Figure 7).

In order to comprehensively evaluate the polishing process under given parameters, this was also realised without an external magnetic field. It was proven that even after twice as long process time, i.e., 10 min, the basic surface parameters remained unchanged, which confirms the necessity of the presence of a magnetic field during the MR polishing process. In other words, a lapping process rather than an MR polishing process was observed in the absence of a magnetic field, when the time of the lapping process to obtain a surface characteristic comparable with the used MR process would be in the order of tens of minutes, or even hours. In addition, the composition of the examined MR slurry is not optimal for the lapping process due to the presence of IP in high concentration, which, due to their lower hardness compared to standard abrasive particles, suffer from their significant forming. This experiment clearly confirmed the necessity of applying a magnetic field during the MR polishing process, when the created chain-like structures are sufficiently (Figure 2 and Figure 3) stiff (no forming of IP occurs) to keep the abrasive particles in contact with the workpiece, thus ensuring the efficiency of the entire process.

An experiment with the original MRF, i.e., without the addition of sepiolite (proportionally increased concentrations of other components) was also performed to evaluate the effect of the clay on the MR polishing process. Only negligible improvements of the basic surface parameters still within the 95% confidence interval of the normalized Δ*S*_p_ were observed in this case (10 samples were measured under the same polishing conditions). It is worth noting here that the addition of sepiolite reduced the sedimentation rate of the MR slurry to a quarter of its original value, but did not affect the MR polishing process significantly.

## 3. Materials and Methods

### 3.1. Magnetorheological Slurry Composition

Magnetorheological slurry comprises carrier liquid (Ferofluid; Unimagnet Ltd., the Czech Republic), iron particles (BASF CN-grade, Germany), abrasive particles of Al_2_O_3_ (KeranikCZ Ltd., the Czech Republic), and sepiolite particles (Sigma-Aldrich, MA, USA) used as a thickening material of the slurry. Concentration of individual components of the slurry can be found in Table 1.

The slurry was prepared by mixing all the components and their subsequent thorough ultrasonication in a beaker before each measurement.

### 3.2. Polished Sample Material

Polyamide PA-6 (RAVAVAMID B, M-Base Engineering + Software GmbH) was chosen as a material to be polished. Before processing, PA-6 was firstly dried at 80 °C for 12 h. Then, the test samples of the size 20 × 20 × 2 mm were prepared via injection moulding using an Arburg Allrounder 470 E Golden Electric machine with a nozzle temperature of 245 °C, pressure of 1200 bar, and a cooling time of 30 s. To fix the samples for in the MR surface polishing device, the test samples were subsequently cast into thermosetting resin (GAFORM D30, Dawex Chemical, the Czech Republic).

### 3.3. Investigation of the Magnetorheological Slurry

The size and morphology of the particles were observed by (SEM) Nova NanoSEM 450 (FEI, USA). Magnetic properties of the MR slurry were determined through measuring its magnetisation curve using a Vibrating Sample Magnetometer (LakeShore, USA) within the external magnetic field intensity range of +/−750 kA m^−1^. The MR properties of the slurry were measured in steady shear at 25 °C using rotational rheometer Physica MCR502 (Anton Paar GmbH, Austria) equipped with a magneto-cell Physica MRD 180/1T. Correct reading of true magnetic flux density was ensured by using a Hall probe. Set temperature and its stability of ±0.02 °C was controlled with a Viscotherm VT2 circulator (Anton Paar GmbH, Austria). A parallel-plate measuring system with a diameter of 20 mm and a gap of 1 mm was employed. Shear rate range was selected to cover both low rates required for correct determination of dynamic yield stress, $\tau_{0}$, and high rates which are reached during MR polishing. Sedimentation of MR slurry with and without added clay in ferrofluid was determined using the tensiometer Krüss K100 (KRÜSS GmbH, Germany). In principle, a crater-shaped measuring probe (SH0640) is attached to a balance and at a defined rate moved to the surface of the material under test (7 mm min^−1^ in our case). Once the contact of the probe with the surface is established, the probe is further immersed at a controlled rate (50 mm min^−1^, in our case) into the homogenously-dispersed (proper mixing and sonication for 2 min) MR slurry to a given immersion depth (15 mm, in our case). The measuring probe collects the particles settled and records their weight in time, thus enabling sedimentation rate calculation [37].

### 3.4. Polishing Process

The MR surface polishing process was realised using custom-designed MR equipment for polishing non-metallic materials [38]. This device allows the polishing of flat surfaces with a maximum size of 40 × 40 mm. Two servomotors with electric drive ensure the smoothness and speed control of the polishing process. The size of the gap between the material and the permanent magnet can be set by a mobile mechanism in the range of 0–160 mm with an accuracy of 0.1 mm (Figure 8).

An area of surface properties measurement was marked on the PA-6 sample in order to ensure repeatability of the measurement. Subsequently, the initial depth of the cut and the surface roughness were measured. In the next step, the sample was fixed in the MR device and a given volume (0.12–0.36 mL) of an MR slurry (which was sonicated before use to ensure better dispersion properties [39]) was applied according to the size of the gap between the permanent magnet and the polished sample (tested gaps: 0.5, 1.0, and 1.5 mm). The MR polishing process was performed at 15 rps in the presence of a magnetic field with an intensity ranging from 460 kA/m to 384 kA/m (verified by a teslameter (Magnet-Physik, FH 51, Dr. Steingroever, Germany) with a Hall probe) for 5 min. Afterwards, the depth of cut and surface roughness of the sample were measured. The surface quality was evaluated using an optical 3D scanner (Zygo NewView 8000) with resolution in the z-axis of 0.1 nm.

## 4. Conclusions

A novel MR slurry recipe based on the addition of clay particles, namely sepiolite, has been designed and tested. The advantage of added sepiolite, a material widely available in nature, lies in the substantial reduction in MR slurry’s sedimentation. The sedimentation rate of the novel MR slurry was reduced to a quarter (1.8 to 0.45 mg s^−1^) compared to the original MR slurry without 5 wt.% of sepiolite. The substantial magnetorheological effect of the investigated MR slurry was confirmed by rheological measurements, which showed a pronounced increase of viscosity and yield stress with applied magnetic field. In other words, the presence of a small amount of sepiolite did not affect the formation of internal chain-like structures in the MR slurry when a magnetic field was applied and the contact regions of each particle were not saturated, resulting in the possible use of a wider processing window during the MR polishing process than that used in this study, i.e., the novel MR slurry is highly efficient. New MR slurry composition was verified in practice on the polishing device of our own construction. During testing it was found that the gap between the magnet and the workpiece significantly impacts the final roughness of the test samples made of PA-6. Future follow up research aims to optimize other remaining polishing factors, i.e., wheel speed, time, and slurry composition, to obtain a general technical model for a 3D texture evaluation of MR polishing.

## Figures and Tables

**Figure 1 ijms-23-12187-f001:**
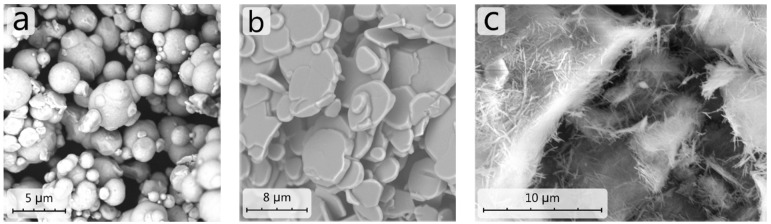
SEM images of iron particles (**a**), Al_2_O_3_ abrasive particles (**b**), and clay mineral sepiolite (**c**).

**Figure 2 ijms-23-12187-f002:**
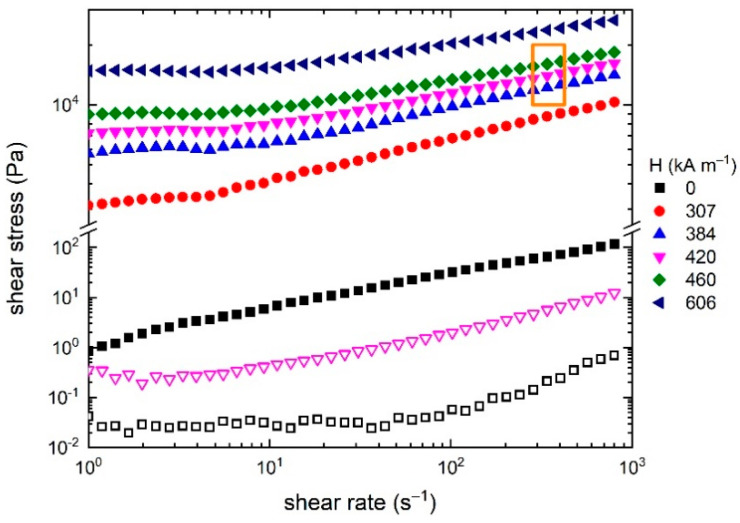
Flow curves at various magnetic field intensity; processing window marked in orange. Full symbols: MR slurry; empty symbols: carrier liquid (ferrofluid) only.

**Figure 3 ijms-23-12187-f003:**
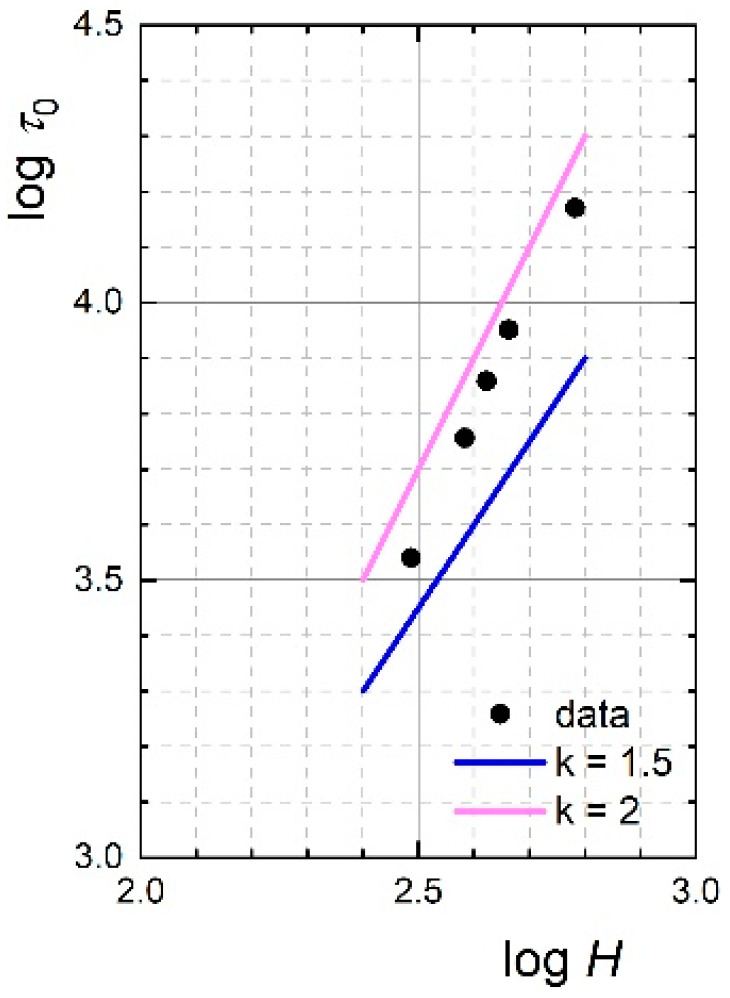
Dependence of the dynamic yield stress on the applied magnetic field for the MR slurry.

**Figure 4 ijms-23-12187-f004:**
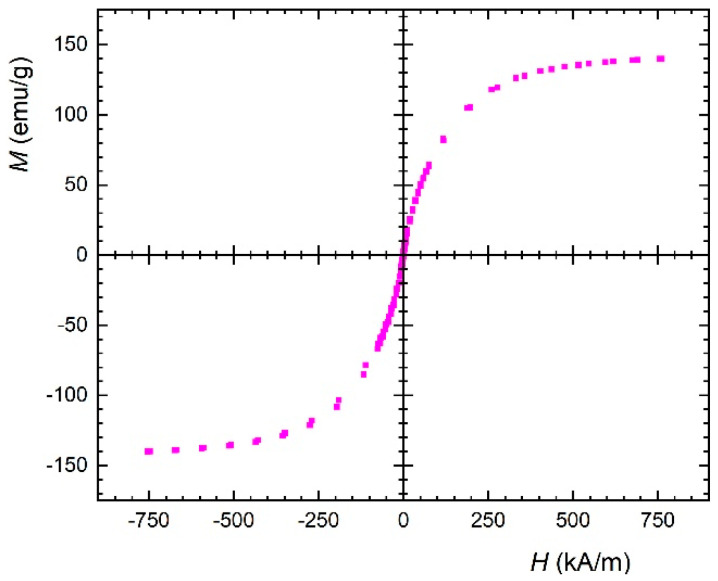
Magnetisation curve of the MR slurry.

**Figure 5 ijms-23-12187-f005:**
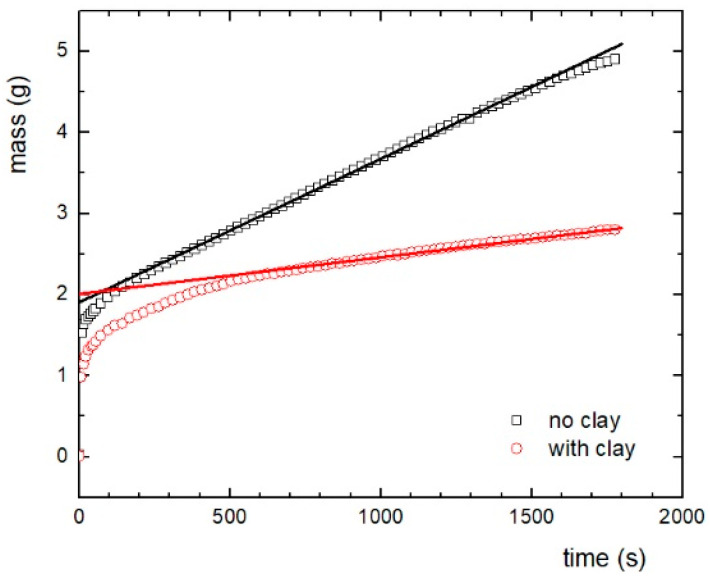
Sedimentation rate of MR slurry with and without clay added. The lines represent an approximate linear fashion with time to quantify the observed sedimentation rate.

**Figure 6 ijms-23-12187-f006:**
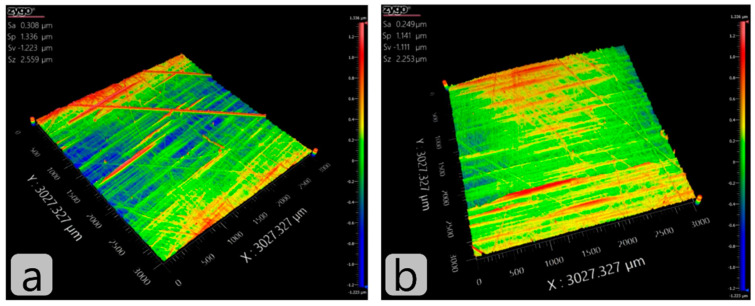
Sample of PA-6 before (**a**) and after (**b**) the process of MR polishing with 15 rps, 5 min polishing time, and magnetic field intensity of 420 kA/m.

**Figure 7 ijms-23-12187-f007:**
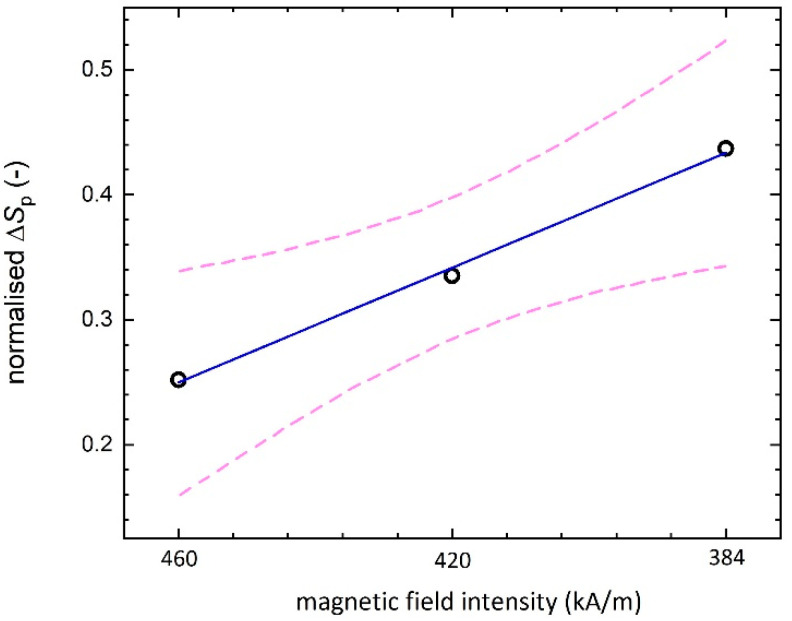
Polishing efficiency versus magnetic field applied; linear regression (full blue line) and its 95% confidence interval (dotted pink lines) are marked.

**Figure 8 ijms-23-12187-f008:**
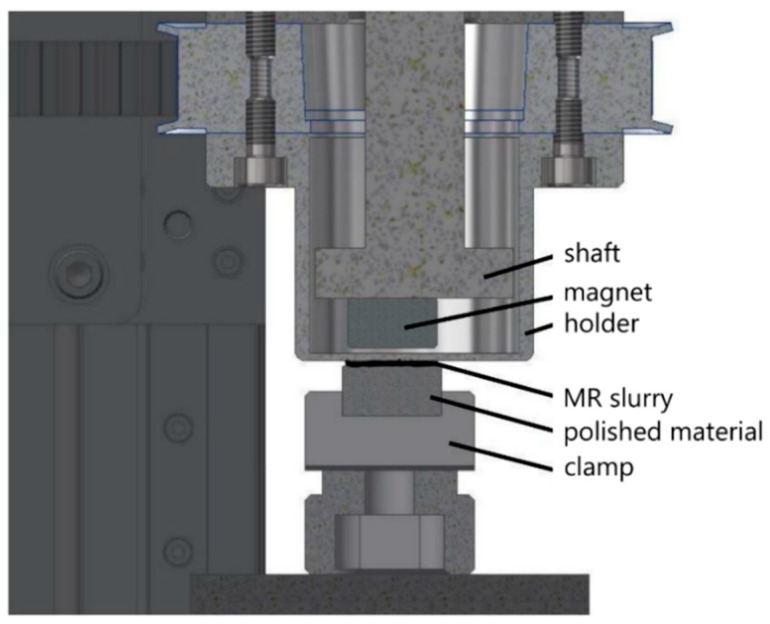
Polishing device used.

**Table 1 ijms-23-12187-t001:** Composition of the MR slurry.

Component	Concentration (wt.%)	Function
iron	55	magnetic part
carrier liquid	30	provides liquid character
Al_2_O_3_	10	abrasion
clay	5	lowers sedimentation

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
