# Peer review of "Iron-Sepiolite High-Performance Magnetorheological Polishing Fluid with Reduced Sedimentation"

_ijms, 2022, doi:10.3390/ijms232012187_

Round 1
Reviewer 1 Report
The manuscript presented by Milde et al. shows an iron-sepiolite composite mixture for improving the polishing properties of fluid-like polishing agents. The information presented in the article is worth publishing and is of interest to the journal readers. However, some amendments are recommended before the article is ready to be accepted.
1) The abstract lacks information about the novelty of the work and how it correlates with previous work. Please, add the information to increase the impact of the study.
2) In the introduction, line 2: what is meant by "high surface accuracy"?
3) The manuscript lacks references in several parts. For instance, entire sections in the introduction do not have a reference or only have 1 or 2. References should support each piece of information. I suggest the authors fix this.
4) In general, the introduction is very long. I suggest reducing the length.
5) Which dangerous solvents are referred to on Page 3, line 1?
6) What is the advantage of this method compared to methods already commercially used today?
7) Page 3, section 2.1., line 4: what is meant by "fibers fibrils"? How are these structures formed?
8) The SEM images of the iron particles and abrasive particles are presented. Why not the SEM images of the mixture used for the study?
9) In rheology, why not perform a study of the viscosity over time? Because the paste has been sensible to sedimentation before, this can also give a hint on the improvement of this property for this suggested paste.
10) How is this method better/improved compared to other methods? The bases for this argument should be clarified.
11) The different components were mixed to prepare the slurry. Why not try the slurry mixing the individual components? This can help understand the mechanisms and know each component's essential role in the final slurry.
Author Response
General remark:
The manuscript presented by Milde et al. shows an iron-sepiolite composite mixture for improving the polishing properties of fluid-like polishing agents. The information presented in the article is worth publishing and is of interest to the journal readers. However, some amendments are recommended before the article is ready to be accepted.
- Answer: We would like to thank the Reviewer for this kind overall evaluation of our manuscript.
Points to be addressed:
- The abstract lacks information about the novelty of the work and how it correlates with previous work. Please, add the information to increase the impact of the study.
- Answer: According to the Reviewer’s recommendation, the last sentence of the Abstract is added to better illustrate the importance and novelty of our work (please see the revised text).
- In the introduction, line 2: what is meant by "high surface accuracy"?
- Answer: The term “high surface accuracy” was intended to present a high quality surface in terms of four basic surface parameters used also in this study, i.e. an average arithmetic deviation of the investigated profile Sa, maximum peak height Sp, maximum valley height Sv, sum of the maximum peak and maximum valley Sz.
- The manuscript lacks references in several parts. For instance, entire sections in the introduction do not have a reference or only have 1 or 2. References should support each piece of information. I suggest the authors fix this.
- Answer: We highly acknowledge the Reviewer for alerting us to referencing all necessary parts of the text to relevant literature sources which definitely support the theoretical background, and increase scientific quality of the manuscript (please see the revised manuscript).
- In general, the introduction is very long. I suggest reducing the length.
- Answer: The length of the Introduction section was shortened to approximately 90% of the original length when we followed three main movements of a standard Introduction section, i.e. (i) establishing a research territory (about 55 % of the current length), establishing a niche (about 35 %), and (iii) occupying the niche (about 10 %).
- Which dangerous solvents are referred to on Page 3, line 1?
- Answer: Until now, only research groups of Prof. Choi (e.g. DOI 10.1007/s00396-009-2013-4), Prof. Gordaninejad (e.g. DOI 10.1002/app.38199), and our group (e.g. DOI 1021/acs.macromol.6b02041) studied the use of surface-initiated atom transfer radical polymerization (ATRP) for the synthesis of core-shell structured magnetic particles useable as a dispersed phase in MR fluids with improved stability properties. This fact shows a complex character of ATRP hindering its scale up into practice due to several reasons. The studies were carried out more to expand fundamental knowledge in the field of possible modifications of magnetic particles for MR systems. Regarding the Reviewer's question, it is possible to list, for example, tetrahydrofuran or toluene as dangerous solvents used in ATRP.
- What is the advantage of this method compared to methods already commercially used today?
- Answer: Commercially used MR fluids as well as MR slurries are primarily based on the use of thickening agents preventing the sedimentation of magnetic particles in the system. However, these additives reduce the magnetic character of the entire system, and more magnetic particles must be used for a sufficient MR effect. A higher concentration of systems increases its viscosity in the inactive state (without an applied magnetic field), which can be a limiting factor in some applications, including MR polishing due to the need to transport MR slurry. As we proved in our study, the addition of only a small amount of clay particles, namely sepiolite (a material commonly available in nature), did not affect the magnetization of the system.
- Page 3, section 2.1., line 4: what is meant by "fibers fibrils"? How are these structures formed?
- Answer: We appreciate Reviewer’s comment and we have changed the morphology description of sepiolite particles in the revised version.
- The SEM images of the iron particles and abrasive particles are presented. Why not the SEM images of the mixture used for the study?
- Answer: We decided not to present the SEM image of the mixture due to much higher amount of iron particles in the mixture.
- In rheology, why not perform a study of the viscosity over time? Because the paste has been sensible to sedimentation before, this can also give a hint on the improvement of this property for this suggested paste.
- Answer: We agree with the Reviewer regarding the possibility of evaluating the long-term stability of the presented MR slurry via rheology. The most commonly used method to evaluate the sedimentation stability of MR fluids is visual observation, which could not be performed because the carrier liquid was composed of ferrofluid, which is also opaque, and it would be imprecise to evaluate the boundary between suspension and oil-rich phase. This is why we used a tensiometric analysis, which recorded the sedimentation rate directly without the need for further conversion as would have to be done from monitoring the viscosity over time.
- How is this method better/improved compared to other methods? The bases for this argument should be clarified.
- Answer: We are not sure what the Reviewer meant by “this method”. If related to the previous question, an explanation of the easier interpretation of the sedimentation rate using the standard sedimentation method is given in the previous answer. If “this method” stands generally for improved long-term stability of the presented MR slurry, the addition of clay mineral in only a small amount creates steric barriers preventing the settling of other components of the MR slurry, while its magnetization and viscosity in the inactive state are changed only negligibly, which, together with the benefit of the clay mineral sepiolite consisting in its availability from natural resources, represents indisputable benefits compared to the methods presented in the Introduction section of the revised version of the manuscript.
- The different components were mixed to prepare the slurry. Why not try the slurry mixing the individual components? This can help understand the mechanisms and know each component's essential role in the final slurry.
- Answer: The components were selected from the literature to represent the most commonly used ones. The essential role of each component can be briefly described as follows: (i) carbonyl iron particles are used as a magnetic agent in the MR slurry responsible for its response to the applied external magnetic field (forming an internal chain-like structure), (ii) ferrofluid is used to give the MR slurry its liquid character, (iii) Al2O3 is used as an abrasive which is pressed on the surface of the workpiece by the internal chain-like structure in the active state, and (iv) sepiolite is used as a stearic barrier preventing the settling of magnetic particles and abrasive particles. The MR slurry would not work without any of the listed components.
Reviewer 2 Report
1. The composition of MR slurry in 3.1 should be presented in Section 2, followed by its field-dependent material characterization: complex moduli, particle-size distribution, surfacr hardness, magnetic propeties, etc.
2. In the synthesis of MR slurry, how to determine concentration % of each component? This can affect the polishing significantly.
3. The polishing results should be presented with and without the magnetic field. In addition, a comparison should be made between the orginal MRF and the proposed MR slurry. In this case, the authros should clearly address on the reason the relationship between the sedimentation and polshing effect. This is the most important finding to be clearly advocated by the authors.
4. How to measure the sedimentation? Thefigure or schematic diagram is required. In addition, The polishing measurement shown in Figure 8 is difficult to understand.
5. How to control the magnetic field for the polishing ?
6. The authors should survey and cite more references related to the articles on the polishing using ER Fluid and MR Fluid. There are many works on this topic published recently.
IN conclusion, this paper should be rejected on the basis on the above comments. However, if the authors can answer to the comments and reflect it to the revised version, i will be happy to review again.
Author Response
Points to be addressed:
- The composition of MR slurry in 3.1 should be presented in Section 2, followed by its field-dependent material characterization: complex moduli, particle-size distribution, surfacr hardness, magnetic propeties, etc.
- Answer: According to the Reviewer’s recommendation, Table 1 presenting the composition of the investigated MR slurry was moved before discussing its properties in the revised version. The sections of the manuscript are ordered according to the journal’s requirements (see Instructions for Authors), but we agree that the system should be defined before its observation.
- In the synthesis of MR slurry, how to determine concentration % of each component? This can affect the polishing significantly.
- Answer: The concentration of individual components is given in wt. %. We definitely agree with the Reviewer that the concentration of the components affects the polishing efficiency, but that was not our primary focus. The purpose of this study is to prepare an MR slurry with prolonged long-term stability. The sedimentation of the main components in the new MR slurry (magnetic particles and abrasive particles) was significantly suppressed by the addition of clay sepiolite in a small amount, moreover, the ability to polish the polymer surface was also demonstrated. The individual components for MR slurry were selected from the literature in such a way to represent the most used ones (in terms of material and concentration).
- The polishing results should be presented with and without the magnetic field. In addition, a comparison should be made between the orginal MRF and the proposed MR slurry. In this case, the authros should clearly address on the reason the relationship between the sedimentation and polshing effect. This is the most important finding to be clearly advocated by the authors.
- Answer: We carefully considered this remark. Application of a magnetic field is one of the necessary prerequisites for an effective MR polishing process. Without a magnetic field, chain-like structures are not formed and abrasive particles are not pressed onto the surface of the workpiece, so polishing does not occur. Hypothetically, some polishing could occur after a very long time, but it would not reach the qualities achieved with a magnetic field. We also performed an experiment with the original MRF, i.e. without the addition of sepiolite (proportionally increased concentrations of other components) and only negligible improvements of the basic surface parameters still within the 95% confidence interval of the normalized ΔSp were observed (10 samples were measured for the same polishing conditions). It is worth noting here that the addition of sepiolite reduced the sedimentation rate of the MR slurry to a quarter of its original value.
- How to measure the sedimentation? Thefigure or schematic diagram is required. In addition, The polishing measurement shown in Figure 8 is difficult to understand.
- Answer: The sedimentation measurement is a standard method for the tensiometer Krüss K100 (KRÜSS GmbH, Germany) when the measuring probe SH0640 is used (https://www.kruss-scientific.com/en/products-services/accessories/sh0640). This analytical device is used not only for MR suspensions.
The principle of the MR polishing device is discussed in more detail in our previous work (DOI 10.1088/1757-899X/726/1/012009). Briefly, the workpiece is clamped in a fixture and a given amount of MR slurry is placed on its surface. Then the permanent magnet is moved down to a given distance from the workpiece surface to guarantee the required magnetic field strength. Then the permanent magnet starts rotating at a given speed and for a given time. We believe that Figure 8 together with the relevant text in the revised version are sufficient to illustrate the functionality of the MR polishing device. This device, which is not as complex as others used for various surface shapes or commercial ones, was used precisely to demonstrate the applicability of proposed MR slurry in the MR polishing process.
- How to control the magnetic field for the polishing ?
- Answer: The magnetic field was controlled by the distance of the permanent magnet from the workpiece surface and was verified by a teslameter with a Hall probe. What is also important and little reflected in the studies focused on the design of new MR slurries for the MR polishing process is the fact that the MR characterization (rheology) was performed under the same conditions as the MR polishing process took place, i.e. within the processing window.
- The authors should survey and cite more references related to the articles on the polishing using ER Fluid and MR Fluid. There are many works on this topic published recently.
- Answer: We thank the Reviewer for this valuable comment. In the revised version, we have added recent (not older than 20 years) references related to the sedimentation problem in MR slurries, when the search keywords in the Web of Science database were used as follows: magnetorheol* polish* sedimentation. It is worth noting that not all the references addressed the validation of their MR slurries in the MR polishing process as performed in our study.
General remark:
IN conclusion, this paper should be rejected on the basis on the above comments. However, if the authors can answer to the comments and reflect it to the revised version, i will be happy to review again.
- Answer: We thank the Reviewer for her/his feedback and valuable advice to improve the manuscript. We have addressed all the remarks above in the best possible way, and the manuscript was edited accordingly. We hope that the revision will be considered satisfactory.
Reviewer 3 Report
The paper refers to magnetorheological polishing fluids and thus it fits the scope of the journal.
The paper need some improvements:
1. Chapter 3.1. How the composition was selected? What were criteria for selection of particular components and their ratios? Did the Authors performed any optimisation?
2. Did the Authors compare their materials with similar compostions (their own or presented in publications by other groups)?
3. Chapter about Materials and Methods should be before Results not after. Details about equipment should be given.
4. Conlusions section is too short and general. There is no numerical parameters?
5. What are key advantages of developed material in comparison withi similar ones (with paramaters)? Thus, what are possible applications (in reference to parameters not only general)?
Author Response
General remark:
The paper refers to magnetorheological polishing fluids and thus it fits the scope of the journal.
The paper need some improvements:
- Answer: We thank the Reviewer’s for her/his advice and positive feedback on this manuscript. Minor comments (as below) were carefully addressed, and the suggestions were taken into deep consideration during the revision.
Points to be addressed:
- Chapter 3.1. How the composition was selected? What were criteria for selection of particular components and their ratios? Did the Authors performed any optimisation?
- Answer: We optimized the composition of our MR slurry (in the revised version Table 1 moved to chapter 2.1.) based on a literature search with the inclusion of the most commonly used components of slurries used for the MR polishing process. Carbonyl iron particles are used as magnetic particles due to their soft magnetic character and availability; the widely used Al2O3 is one of the most frequently used abrasive particles in MR slurry; and ferrofluid was used as a dispersion medium, also due to its partial magnetic character. The proposed additive suppressing sedimentation, natural clay in the form of sepiolite, was again chosen due to its availability. The concentrations of magnetic particles and abrasive were chosen again within the standard range.
- Did the Authors compare their materials with similar compostions (their own or presented in publications by other groups)?
- Answer: As partially stated in the previous answer and also in the answer to respected Reviewer #2, our goal was not to develop MR slurry with the highest efficiency, as is the case with the majority of studies focused on the development of MR slurries for MR polishing process, but rather to use the most universal MR composition possible The sedimentation stability of the new MR slurry has been significantly increased, which can be used in equipment used for the MR polishing process, where the MR slurry is located in a reservoir, in which it can easily sediment if the MR slurry is not used for a long time.
- Chapter about Materials and Methods should be before Results not after. Details about equipment should be given.
- Answer: We completely understand the Reviewer's attitude regarding the organization of the manuscript, but the sections of the manuscript are ordered according to the journal’s requirements (see Instructions for Authors).
- Conlusions section is too short and general. There is no numerical parameters?
- Answer: Following the Reviewer’s advice, the most important results are discussed in more details in Conclusions section of the revised manuscript.
- What are key advantages of developed material in comparison withi similar ones (with paramaters)? Thus, what are possible applications (in reference to parameters not only general)?
- Answer: As discussed above, the key advantage of the developed MR slurry is its prolonged sedimentation stability, which was achieved by only slight modification of the composition of the standard MR slurry by addition of naturally available clay sepiolite, i.e. it was not necessary to use the complex procedures described in recent literature dealing with the development of sedimentation-stable MR slurry. Furthermore it was proven that the new MR slurry possesses a high MR effect and, based on its main composition taken from most frequently used MR slurry, also standard polishing characteristics.
Round 2
Reviewer 1 Report
The revised manuscript has answered some of the points raised by the reviewers. However, I did not observe many changes in the manuscript, considering that the decision was major revisions. In addition, many of the reviewers comments were only answered in the response to the reviewers but not on the resubmitted text. I suggest the authors amending this in the new version.
Author Response
We would like to thank the Reviewer for this suggestion and have transferred the comments/answers relevant to the study from the previous Response to Reviewers letter into the manuscript as seen in its revised version.
Reviewer 2 Report
The responses to my comments are not satisfied in the following issues.
1. The polishing results with and without the magnetic field should be compared. Even though, the polishing effect without the magnetic field for a short time is very low, the readers would like to know the characteristics of the polishing materials
2. The authors answered that the sedimentation measurement is a standard method without reflecting this issue to the revised version. This point is required to the readers who are not familiar with MRF sedimentation. In addition, the authors should express test conditions in details.
3. The authors did not show the controllability of the polishing results. The magnetic field can be determined by the distance of the permanent. In fact, the gap of Figure 7 can be converted to the magnetic field.
4. The authors asserted that the main contribution of this work is to use MR polishing slurry instead of conventional MRF. Then, a comparative result between the proposed and conventional method should be undertaken to demonstrate some benefits of the proposed method or salient characteristics those are occurred in the conventional MRF. This is the most critical point to meet the motivation of this work.
Author Response
- The polishing results with and without the magnetic field should be compared. Even though, the polishing effect without the magnetic field for a short time is very low, the readers would like to know the characteristics of the polishing materials
- Answer: According to the Reviewer’s recommendation, we performed another experiment without an applied magnetic field. The results agreed with our previous assumptions and the necessity of the magnetic field was proven. We commented on this experiment and the results in detail in the revised version of the manuscript.
- The authors answered that the sedimentation measurement is a standard method without reflecting this issue to the revised version. This point is required to the readers who are not familiar with MRF sedimentation. In addition, the authors should express test conditions in details.
- Answer: The revised version includes a discussion of the need to use a different method for evaluating the sedimentation of dispersed particles than is most frequently used in MRFs (visual observation), as well as the specific conditions for the experiment that have already been used for MRFs in the past, as it is supported by relevant reference.
- The authors did not show the controllability of the polishing results. The magnetic field can be determined by the distance of the permanent. In fact, the gap of Figure 7 can be converted to the magnetic field.
- Answer: We thank the Reviewer for this valuable comment. In the revised version, we changed the parameter responsible for the controllability of the polishing results from “gap” to “magnetic field intensity”. This is also reflected in Figure 7.
- The authors asserted that the main contribution of this work is to use MR polishing slurry instead of conventional MRF. Then, a comparative result between the proposed and conventional method should be undertaken to demonstrate some benefits of the proposed method or salient characteristics those are occurred in the conventional MRF. This is the most critical point to meet the motivation of this work.
- Answer: In general, the MR polishing process is based on the formation of chain-like internal structures from iron particles that push the abrasive particles into contact with the workpiece surface. In other words, the stiffness of this “brush” is adjustable by changing the applied magnetic field, but the polishing is realized only by the abrasive particles. The situation without a magnetic field is discussed in the first answer and in the revised version of the manuscript. Both types of particles, i.e. magnetic particles and abrasive particles, are necessary for the MR polishing process. Standard MRF is free of abrasive particles and considering the requirement to have more than 4 HRC harder abrasive particles than the HRC of the workpiece, conventional MRF is not suitable for the MR polishing process (also discussed in the revised version of the manuscript). In the revised version, a comparison between the polishing results for the standard MR slurry (composed based on the literature search) and its improved variant (containing sepiolite) by means of improved sedimentation stability is also discussed.
Reviewer 3 Report
The paper fits the scope of the journal. Its scientific soundness and novelty are moderate. It is rather case study. However, the paper has sufficient qyality and can be considered for publication.
Author Response
We would like to thank the Reviewer for this kind overall evaluation of our manuscript.
Round 3
Reviewer 2 Report
In the revised version, the following comments have not been properly answered.
1. It is seen from Figue 7 that the polishing efficiency is increased as the magnetic field intensity is increased. Why? The presentation of SM images of the polishing results at low and high magnetic field intensity will validate this.
2. In the title, we see the relationship between high performance MR polishing and reduced sedimentation. In order to clearly validate this feature, five different MR fluids which have different sedimentation rate should be tested to get the result polyishing versus sedimentation rate.
3. It is seen from Figure 3 that the field-dependent properties of the yield stress. What is the relationship between this and polishing performance?
4. The sedimentation property of MR fluid used in this work should be presented and compared with one of commercial products to understand how slow the sediementation is occurred in the proposed MR fluid.
Author Response
In the revised version, the following comments have not been properly answered.
- Answer: We would like to thank the Reviewer for creating a completely new set of questions to improve the quality of the presented results.
- It is seen from Figue 7 that the polishing efficiency is increased as the magnetic field intensity is increased. Why? The presentation of SM images of the polishing results at low and high magnetic field intensity will validate this.
- Answer: We thank the Reviewer for this valuable comment, which alerted us to a mistake we made when changed the presentation of the x-axis in Figure 7 from “gap” to “magnetic field intensity” in a previous revision. Correctly, the intensity of the magnetic field should have a downward tendency with increasing gap, i.e. the distance of the permanent magnet from the surface of the workpiece. Figure 7 is now correct. We also added a discussion about the effect of the magnetic field intensity on surface quality in revised version of the manuscript when only results for “optimal” magnetic field are presented in Fig. 6.
- In the title, we see the relationship between high performance MR polishing and reduced sedimentation. In order to clearly validate this feature, five different MR fluids which have different sedimentation rate should be tested to get the result polyishing versus sedimentation rate.
- Answer: We discussed this point in the revised version of the manuscript appropriately where we explained the necessity to ensure a constant concentration of particles (magnetic ones for chain-like structures formation and abrasive ones to ensure the polishing of the workpiece) over time with regard to the reproducibility and efficiency of the MR polishing process under the given conditions. In other words, reduced particles concentration will result in less efficient MR polishing.
- It is seen from Figure 3 that the field-dependent properties of the yield stress. What is the relationship between this and polishing performance?
- Answer: The relevant discussion is added to Fig. 3 in the revised version of the manuscript.
- The sedimentation property of MR fluid used in this work should be presented and compared with one of commercial products to understand how slow the sediementation is occurred in the proposed MR fluid.
- Answer: Although we currently do not have commercial MR slurry for comparison reasons, when designing the basic MR slurry used in this study, into which the sepiolite particles were added, we did the literature search on the most commonly individual components (including their standard concentration) presented for MR slurries [1–3], which is also discussed in the manuscript. The MR slurry used in our study is therefore not significantly different from the commonly studied ones. In the revised version of the manuscript, we also compared the sedimentation stability with a similar system, where by using the sepiolite mineral we achieved a higher sedimentation stability (4x compared to the original MR slurry) than the approach mentioned in the literature using core-shell magnetic particles (2x compared to the original MR slurry) [1]. The complexity of our study can be further substantiated by the fact that in similar studies dealing with increasing the sedimentation stability of MR slurry, its usability in the MR polishing process is often not subsequently confirmed [2,4].
[1] Hong, K. P.; Song, K. H.; Cho, M. W.; Kwon, S. H.; Choi, H. J., Magnetorheological properties and polishing characteristics of silica-coated carbonyl iron magnetorheological fluid. Journal of Intelligent Material Systems and Structures 2018, 29, (1), 137-146.
[2] Xu, J. H.; Li, J. Y.; Cao, J. G., Effects of fumed silica weight fraction on rheological properties of magnetorheological polishing fluids. Colloid and Polymer Science 2018, 296, (7), 1145-1156.
[3] Kwon, S. H.; Choi, H. J., Magnetorheology of Xanthan-gum-coated Soft Magnetic Carbonyl Iron Microspheres and Their Polishing Characteristics. Journal of Korean Physical Society 2013, 62, (12), 2118-2122.
[4] Guo, C.; Liu, J.; Li, X. H.; Yang, S. Q., Effect of cavitation bubble on the dispersion of magnetorheological polishing fluid under ultrasonic preparation. Ultrasonics Sonochemistry 2021, 79.
Round 4
Reviewer 2 Report
This paper is now acceptable.